# Electrically reconfigurable terahertz signal processing devices using liquid metal components

Kimberly S. Reichel[1,3], Nicolas Lozada-Smith[1], Ishan D. Joshipura[2], Jianjun Ma [1], Rabi Shrestha[1], Rajind Mendis[1], Michael D. Dickey[2] & Daniel M. Mittleman[1]

Many applications of terahertz (THz) technology require the ability to actively manipulate a free space THz beam. Yet, although there have been many reports on the development of devices for THz signal processing, few of these include the possibility of electrical control of the functionality, and novel ideas are needed for active and reconfigurable THz devices. Here, we introduce a new approach, based on the integration of electrically actuated liquid metal components in THz waveguides. This versatile platform offers many possibilities for control of THz spectral content, wave fron"ts, polarization, and power flow. We demonstrate two illustrative examples: the first active power-splitting switch, and the first channel add–drop filter. We show that both of these devices can be used to electrically switch THz communication signals while preserving the information in a high bit-rate-modulated data stream.

[1] School of Engineering, Brown University, Providence, RI 02912, USA. [2] Department of Chemical and Biomolecular Engineering, North Carolina State University, Raleigh, NC 27695, USA. [3] Present address: NEST, CNR-Istituto Nanoscienze, Piazza San Silvestro 12, Pisa I-56127, Italy. Correspondence and requests for materials should be addressed to D.M.M. (email: daniel_mittleman@brown.edu)

The ongoing explosion in wireless data traffic and the looming rollout of millimeter-wave wireless standards have conspired to move the topic of terahertz (THz) wireless communications to the forefront of research activity[1,2]. As a result, considerable research has been directed toward the demonstration of components for manipulating carrier waves with frequencies near or above 100 GHz. In nearly all cases, however, these have been passive mono-functional devices[3–6], or at best have achieved limited tuning of a single parameter[7–11].

An exciting new possibility is suggested by recent developments in the electrical actuation of liquid metals[12,13]. Metals such as Galinstan, a gallium–indium–tin alloy, remain liquid at room temperature, exhibit high electrical conductivity, and unlike mercury possess low toxicity. In a few cases, liquid metals have been used previously for THz[14,15] and RF[16,17] devices. However, such devices actuate the liquid metal using pneumatics or strong acids, which remove the native oxide that otherwise stabilizes the shape of the metal. Recently, we have demonstrated voltage-actuated reconfigurability[18] achieved by applying a small voltage (<4 V) across an electrolyte surrounding the liquid metal. The use of voltage alleviates the need for bulky pneumatic pumps while the electrolyte creates a "slip layer" that prevents the metal from adhering to surfaces. Since the length scale of movement of the liquid metal ranges from millimeter to centimeter scale, these are ideal for reconfigurable RF and THz devices[19,20], unlike MEMS technology that typically provides a much smaller range of movement[21]. The combination of liquid metals, electrically actuated using suitably chosen electrolytes, with passive metallic waveguides[22–26] provides many new possibilities for THz signal processing devices.

Here, we introduce a new approach for the design of reconfigurable, scalable, and multi-functional devices for signal processing of THz carrier waves. This versatile platform combines electrically actuated liquid metal components[12,13] with passive THz waveguides. We demonstrate two different active devices that are enabled by this approach: the first electrically actuated THz power-splitting switch, and the first active THz channel add–drop filter. We characterize the performance of these devices using broadband THz pulses, and we demonstrate their compatibility with wireless transmission systems by switching a 1 Gb/s data stream on a THz carrier wave[4,27,28].

## Results

**Active T-junction power splitter**. As a first demonstration of this versatile approach, we describe a power splitter where the two output ports can be switched on or off using electrically actuated liquid metal components. This work builds on our earlier report of a tunable power splitter based on a T-junction waveguide, which incorporated a movable septum to tune the splitting ratio between the two output arms[5]. Here, we modify this mechanically tunable device by incorporating liquid metal channels into each of the output arms of the T-junction (see Fig. 1a). These channels consist of glass tubes of rectangular cross-section (200 μm by 2 mm), containing a liquid metal plug in an electrolyte solution. For this device, we use an electrolyte solution consisting of 2 M NaOH in water, which has a large absorption coefficient throughout the THz spectral range (see Supplementary Note 1). The application of a small (~4 V) DC voltage across the electrolyte causes the metal plug to move along the channel, into or

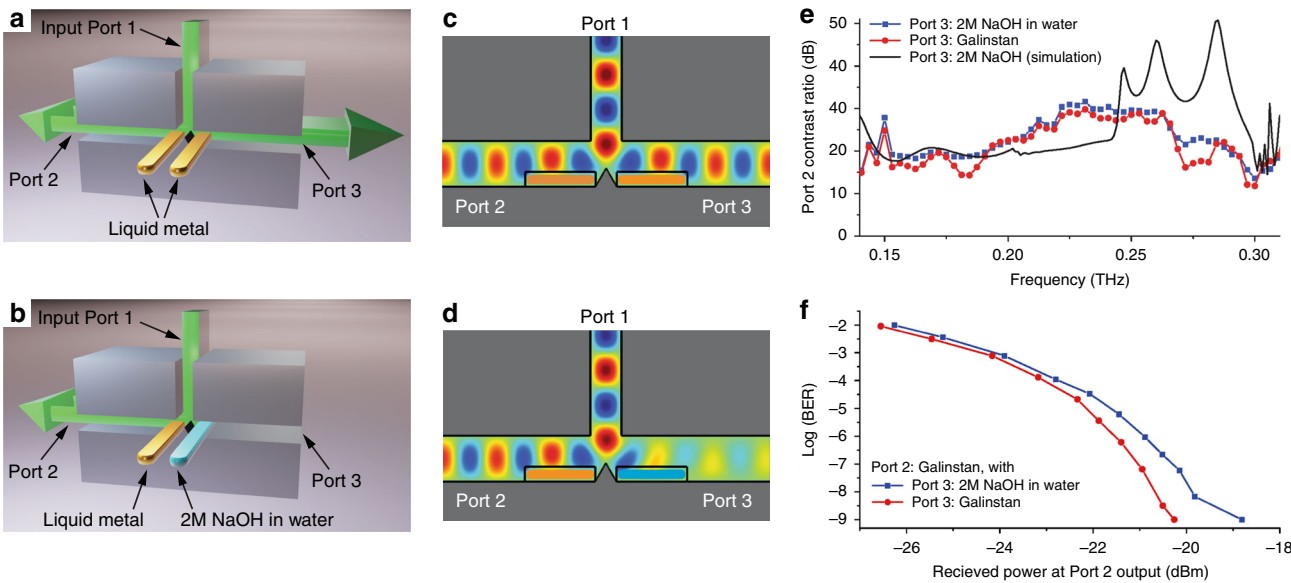

**Fig. 1** A reconfigurable T-junction power switch. A T-junction waveguide with a mechanically repositionable septum can be used as a variable-ratio power splitter. **a**, **b** We incorporate a pair of liquid metal channels into the two output arms of the T. The metal plug can be moved in these channels using a DC voltage. This causes the side wall of the T-junction waveguide section to be either metal (when the metal plug is inserted) or a highly absorbing liquid electrolyte (when the metal plug is withdrawn). Here, the signal emerging to the right (Port 3) is present when the right-hand channel contains metal (**a**), and absent when the metal is withdrawn, leaving only electrolyte (**b**). **c**, **d** Numerical simulations using COMSOL, showing the change in the signal propagating in Port 3 when the liquid metal plug is present (**c**) and when it is replaced with absorbing electrolyte (**d**) on the right side. The channel on the left side (Port 2) has liquid metal in both cases. **e** The measured spectrum of the power ratio (metal plug inserted vs. withdrawn) for the Port 2 output of the T-junction, for the case when Port 3 has liquid metal (red) and electrolyte (blue). A broad spectral region is observed for which the on–off ratio exceeds 20 dB. The two curves are very similar, indicating that the output of Port 2 is unaffected by the state of Port 3. The black solid curve shows the predicted ratio from COMSOL simulations. **f** Characterization of this switch using modulated data at 1 Gb/s, at 200 GHz carrier frequency. The bit error rate for the signal from Port 2 reaches error-free (10$^{-9}$) performance when the metal plug is inserted, for both states of the channel in Port 3. The small difference between these two curves is an indication of a small degree of cross-talk due to a back reflection from the liquid metal plug on the Port 3 side. When the metal plug is removed from Port 2, the signal from that port is much too small to measure a BER

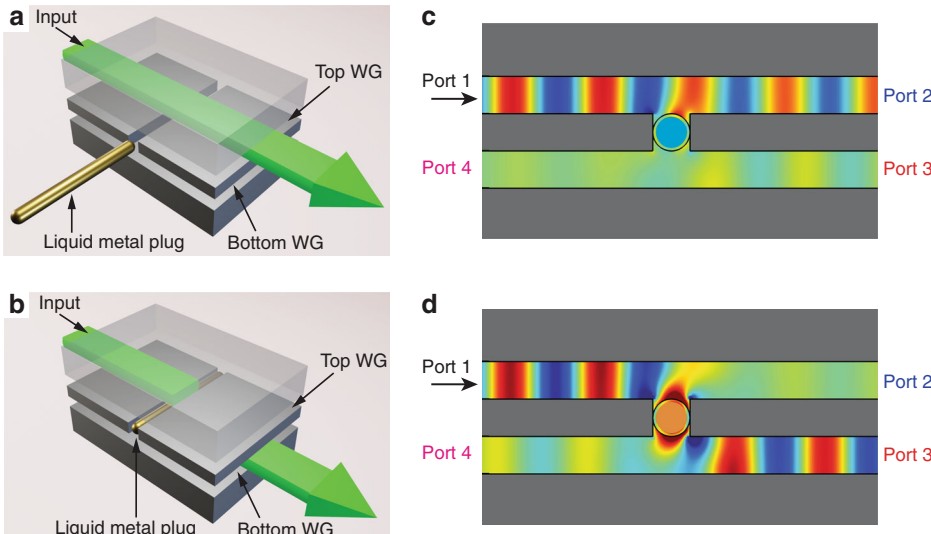

**Fig. 2** A reconfigurable directional coupler based on liquid metal actuation. The directional coupler consists of two vertically stacked PPWGs with a gap in the shared metal slab between them. In this gap, a glass capillary contains an electrolyte solution and a liquid metal plug (shown in gold). By applying a small voltage, the liquid metal plug can be electrically repositioned so that it is out of the beam path (as in **a**, **c**) or in the beam path (as in **b**, **d**), thereby changing the electromagnetic coupling between the two waveguides. **a**, **b** Diagrams of the geometry where the input excites the top waveguide. **c**, **d** Finite-element simulations of the dual waveguide, showing the vertical component of the electric field. The upper waveguide is excited by at Port 1 in the TEM mode, at a frequency of 123 GHz. **a**, **c** When the liquid metal is out of the beam path (so that the capillary is backfilled with electrolyte), most of the THz signal remains in the upper waveguide and exits from Port 2. **b**, **d** When the liquid metal is in the beam path, nearly all of the THz wave couples to the lower waveguide and exits from Port 3

out of the region where the THz beam is propagating, via the continuous electrowetting (CEW) method[18,19,29]. In this fashion, this region of the waveguide can either have a metal sidewall with low propagation loss (when the metal plug is inserted) or a liquid electrolyte sidewall with very high loss (when the plug is withdrawn). We can therefore electrically switch either output port of this power splitter on and off. Figure 1e shows the spectral ratio of the signals emerging from one of the outputs in the "on" and "off" configurations, demonstrating more than 20 dB of contrast over more than 100 GHz of spectral bandwidth.

For any device that is envisioned for use in THz communications systems, it is important to characterize the performance not only using a broadband source (as in Fig. 1e), but also using a modulated data stream, so that a realistic picture of system operation can be obtained[4,27]. For this purpose, we employ a THz communications test bed producing a modulated data stream (amplitude shift keying (ASK) modulation) at several different carrier frequencies, with a data rate of 1 Gb/s[28]. We characterize the device performance by coupling this modulated carrier wave into the input port, and measuring bit error rate (BER) for the signals extracted from the output ports. Figure 1f illustrates typical results, validating the possibility to transmit and switch data through the device with error-free (BER $<10^{-9}$) performance at 200 GHz. Moreover, we see that the data stream is completely blocked when the output port is in the "off" configuration, and that toggling the state of the opposite output port has only a minor effect on the BER (i.e., the two outputs have low cross-talk).

**Coupled waveguides as a channel add–drop filter.** As a second more sophisticated example, we describe a dual-coupled waveguide, which can be used as a channel add–drop filter for spectral selection of multiplexed THz signals[3,4]. We use two vertically stacked parallel plate waveguides (PPWGs) with a gap in the shared wall between them, to allow coupling of the guided THz wave from the top to the bottom waveguide. This gap is filled with one or more capillary tubes. As in the above case of the

power-splitting switch, the liquid metal plug in each capillary is immersed in a liquid electrolyte, and can be moved into or out of the THz beam path with a small DC voltage (see Fig. 2a, b). Similar to the earlier example, we optimize the parameters including electrolyte conductivity and transparency in the THz range, and the geometrical and material parameters of the capillaries. In this case, the electrolyte can be optimized for either high or low THz absorption, depending on the details of the electromagnetic coupling through the gap between the two waveguides (see Supplementary Note 7). This in turn depends on the input waveguide mode, which can be polarized either parallel to the plate surfaces ($TE_1$ mode) or perpendicular to them (TEM mode). For the $TE_1$ case, a low THz absorption is needed, since the coupling requires propagation of the signal through the interior of the electrolyte-filled capillary. This is more challenging, since lowering the THz absorption of the electrolyte also lowers its DC conductivity and therefore the effectiveness of the CEW method. Examples of $TE_1$ mode operation are shown in the Supplementary Note 8; here, we focus on the TEM mode, where the coupling between the two waveguides is mediated almost exclusively by the walls of the capillary, not by the liquid in the interior (thus relaxing the requirements on the THz optical properties of the electrolyte).

Simulations based on the finite-element method (FEM) clarify the coupling mechanism between the two waveguides. Figure 2 illustrates an example for a particular waveguide geometry and for a TEM mode[22] propagating in the upper waveguide. Here, if the liquid metal plug is withdrawn from the beam path (so that the capillary is filled only with electrolyte), the coupling between waveguides is minimized, and the incident THz wave primarily remains in the top waveguide and exits via Port 2 (Fig. 2a, c). In contrast, if the plug of liquid metal is inserted into the beam path, the effective gap between the waveguides changes to two narrow slits (defined by the sidewalls of the glass capillary). As a result, a strong and spectrally narrow resonance arises, which permits transmission of nearly 100% of the THz energy through the capillary walls to the bottom waveguide,

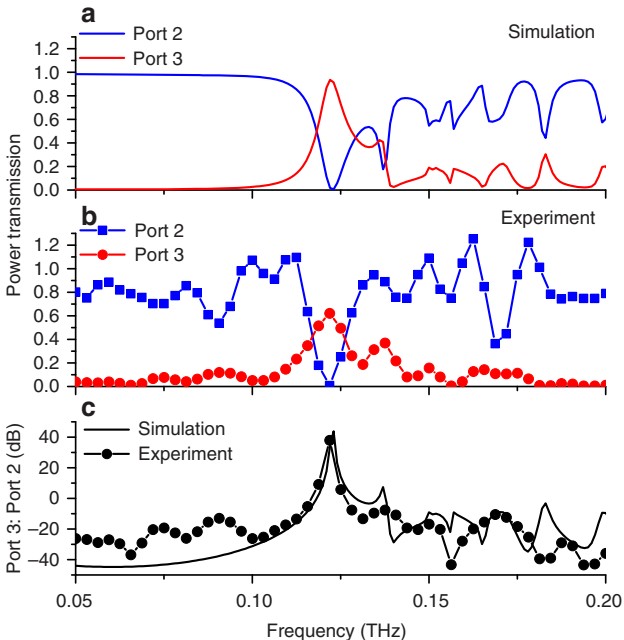

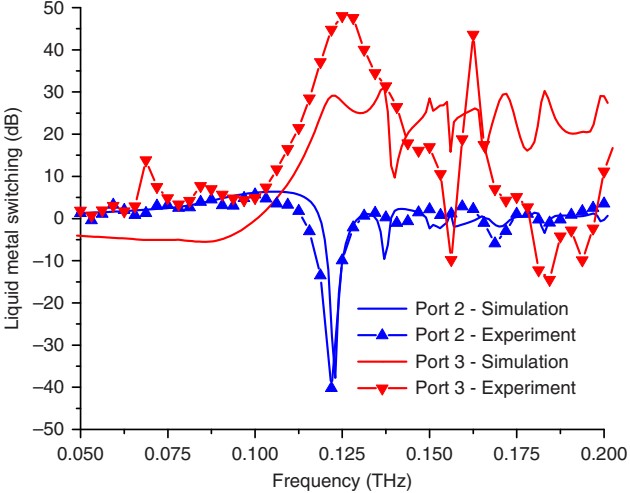

**Fig. 4** Switching ratios for the single capillary filter. The ratio of the two configurations with the liquid metal in the beam path versus the liquid metal out of the beam path, both simulated (solid lines) and measured (symbols). At the resonance at 123 GHz, Port 2 (blue curves) shows a strong dip of −40 dB and Port 3 (red curves) shows a peak of +50 dB

**Fig. 3** Characterization of a single capillary filter. **a** FEM simulations and **b** experimental results for the case when the metal plug is in the beam path. A narrow resonant peak is observed at 123 GHz, as this frequency is coupled to the lower waveguide and emerges through Port 3 (red curves), with a corresponding dip in the signal emerging from Port 2 (blue curves). **c** The isolation ratio (Port 3/Port 2) reaches nearly +40 dB at the resonance frequency

where it exits via Port 3 (Fig. 2b, d). Thus, a signal propagating at the resonant frequency is extracted from the top waveguide, enabling a channel add–drop functionality.

**Experimental characterization using broadband THz pulses**. We first study a single capillary device, using broadband THz pulses from a time-domain spectrometer. We characterize the performance by measuring the signal emerging from either Port 2 or Port 3 of the waveguide, as defined in Fig. 2. As a reference measurement, we replace the shared central metal waveguide plate with a solid metal plate that has no gap between the waveguides. The plate separation is 1 mm for both waveguides. We use a capillary with 1 mm outer diameter and 0.8 mm inner diameter (i.e., 0.1 mm wall thickness), containing a Galinstan plug of 8 mm length. Using a DC offset of 0.1–1 V on top of an AC square wave of 2 $V_{pp}$ and 2 Hz, the metal plug can be selectively repositioned along the capillary. The DC voltage determines the displacement of the liquid metal plug, while the AC component minimizes the splitting of the plug[19]. The capillary is connected to the voltage source via platinum wires placed either directly inside the ends of the filled capillary tube or into reservoirs located at either end. Since transparency of the liquid electrolyte is not required for TEM mode operation (see Fig. 2), we use a similar electrolyte solution to that employed in the T-junction device described above.

We characterize the electromagnetic behavior when the plug is in the beam path and when it is withdrawn. Figure 3 shows the results when the plug is in the beam path, so that the resonant component of the broadband THz signal in the top waveguide can couple to the bottom waveguide through the glass capillary walls, as anticipated by Fig. 2b, d. We define the isolation of the add–drop filter as the ratio of the power transmission from Port 3 to Port 2 when the plug is in the beam path (Fig. 3c). On resonance, we measure nearly 40 dB of isolation between the two ports, in excellent agreement with simulation.

A key attribute of devices based on this design concept is their reconfigurability. For the add–drop filter, a particular frequency channel can be steered to one or the other waveguide by repositioning the liquid metal. To characterize this performance, we compare the output of the two ports of the device in the two liquid metal states (i.e., in the beam path or not). Figure 4 shows the ratio of the outputs of the two ports. At the resonance frequency of 123 GHz, Port 3 shows a ratio of nearly +50 dB between liquid metal states, while the Port 2 ratio is less than −40 dB. These dramatic swings in the transmitted power between the two states of the device, achieved with just a few volts applied, are substantially larger than on−off ratios for other electrically actuated THz signal processing components[7,8,21,30]. We also demonstrate dynamic operation by using an AC voltage to periodically vary the position of the liquid metal plug. At frequencies up to a few Hz, the performance is equivalent to that of the DC measurements shown in Fig. 4 (see Supplementary Notes 2, 5, and 8).

Another important attribute of the approach is scalability, as a device can incorporate many liquid metal components. Figure 5 illustrates similar measurements on a second add–drop filter, this time with two glass capillaries in parallel separated by a 1 mm solid metal spacer and with each containing an independently controlled liquid metal plug. In this device, we observe that the single transmission resonance splits into two resonant peaks, which are in the high transmission state when both metal plugs are in the beam path. Figure 5 shows the experimental characterization, again in good agreement with simulation. We find nearly 30 dB of isolation for both of the dropped channels. From FEM simulations, we find that, when the number of capillaries is increased, more resonances are present, and they are narrower (see Supplementary Note 3).

**Experimental characterization using a data stream**. As above, it is important to show that the device can realistically be used in a communications context, by testing it using a modulated THz data stream. Here, we use a three-capillary device with larger capillary tubes (1.2 mm outer diameter and 1.0 mm inner diameter) for a lower resonance frequency near 100 GHz, to coincide with one of the discrete frequencies at which our communications

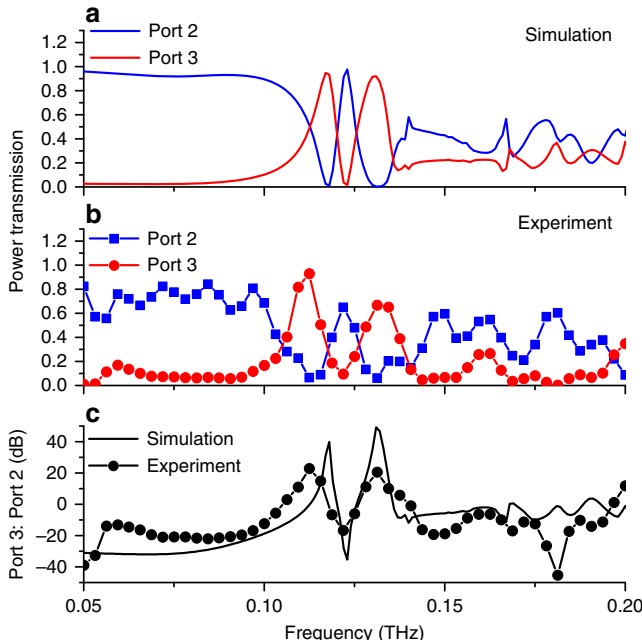

**Fig. 5** Characterization of a two-capillary filter. **a** FEM simulations and **b** experimental data for two 1 mm outer diameter glass capillaries separated by a 1 mm square metallic rod spacer where both capillaries contain a liquid metal plug in the beam path. In this case, there are multiple resonant features. Two narrow resonant peaks can be seen at 116 GHz and 130 GHz in the signals emerging from Port 3 (red curves), while a resonant peak is observed at 123 GHz emerging from Port 2 (blue curves). **c** The ratio of Port 3 to Port 2 reveals an isolation ratio of nearly +40 dB at the two outer resonances, and nearly −40 dB at the middle one

test bed operates. In Fig. 6, we compare two of the eight possible configurations: the 000 configuration (liquid metal plugs in all three capillaries are in the beam path) and the 101 configuration (only the center capillary's plug is in the beam path). As in Fig. 1, the results validate the possibility to transmit and switch data through the filter with error-free (BER <10$^{-9}$) performance, at very reasonable input power levels. When the device is in the 000 configuration, an extra 9 dB of input power are required to achieve error-free threshold in Port 2, compared to Port 3. Conversely, in the 101 configuration, an extra 6 dB of input power are required to reach error-free performance in Port 3, compared to Port 2. These numbers are somewhat diminished from the larger ratios seen in Figs. 4 and 5, because of a small mismatch between the resonant frequency of this filter (at 90 GHz) and the (fixed, not tunable) carrier frequency of our THz communications test bed (at 98 GHz) (see Supplementary Note 6). The filter frequencies can be tuned through a more precise optimization of the filter's geometrical parameters.

In conclusion, we have demonstrated a new design approach for electrically reconfigurable THz signal processing components. As a first demonstration, we have characterized a power-splitting switch and a channel add–drop filter. Both devices are electrically reconfigurable, with superior isolation and switching performance, and the capability for switching signals that are modulated at high data rate. This work, to the best of our knowledge, demonstrates the first dynamic add–drop filter in the THz range. Electrically actuated liquid metals such as these have already been shown to operate at frequencies of up to a few tens of Hz, which is more than sufficient for numerous important device functionalities. Building on this platform, we can envision many different THz components, such as for wavefront shaping and beam steering.

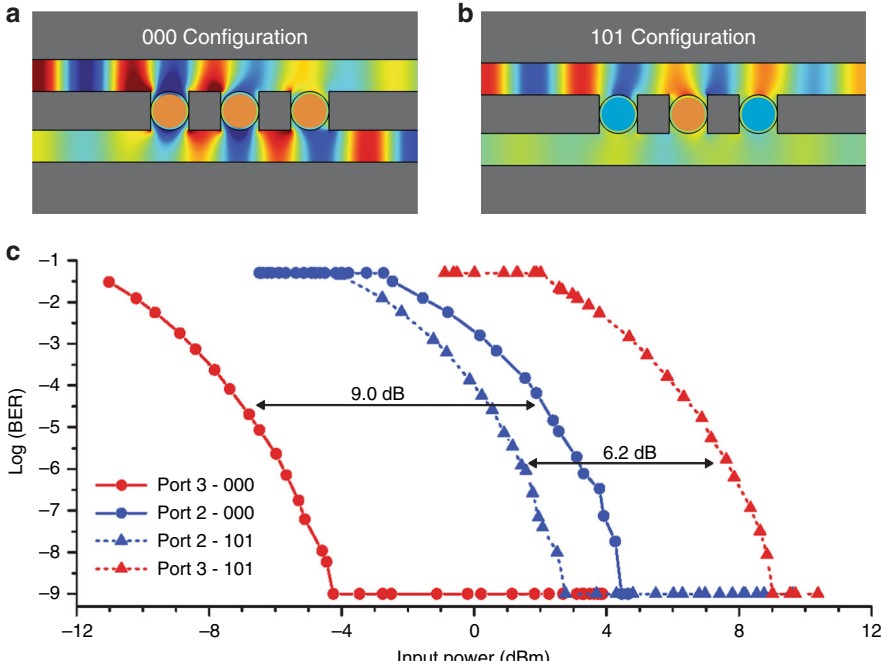

**Fig. 6** Switching a 1 Gb/s data stream using a three-capillary filter. The bit error rate (BER) for two different configurations of a device with three 1.2 mm outer diameter glass capillaries separated by 1 mm metallic rod spacers, measured as a function of power at the input port (Port 1) of the filter. **a** The 000 configuration corresponds to all three capillaries filled with liquid metal. **b** The 101 configuration corresponds only the central capillary filled with liquid metal. **c** Error-free performance (BER <10$^{-9}$) can be achieved in all configurations. When the device is configured to direct the signal out of Port 2 (i.e., the 101 configuration, dashed lines with triangles), error-free performance is much more easily obtained from Port 2 (power requirement reduced by ~6 dB). In contrast, when the device is configured to direct the signal to Port 3 (the 000 configuration, solid curves with circles), ~9 dB less power is required to achieve error-free performance at Port 3

## Methods

**Experiments**. For both the pulsed and modulated data stream experiments, we illuminate Port 1 of the top waveguide with a Gaussian beam polarized perpendicular with respect to the waveguide plates to excite the TEM waveguide mode or parallel with respect to the waveguide plates to excite the $TE_1$ mode. Since the data stream experiments are continuous wave, the waveguide is slightly misaligned from normal with respect to the input beam to avoid back reflections that would negatively interfere with the data transmission.

**Simulations**. We perform finite-element method (FEM) simulations of our device, both in 2D and in 3D. We treat the metal plates and the liquid metal plug as perfect electric conductors (PEC), which is a reasonable approximation in the THz range since the ohmic losses are negligible for these short propagation distances. For the glass capillaries, we use the dielectric values for quartz glass as found in the literature[31]. To accurately model the conductive fluids, we perform time-domain spectroscopy measurements with a variable path length liquid cell[32] to obtain the refractive index and absorption coefficients of various electrolyte mixtures (see Supplementary Note 1). From the 3D simulations, we find that the liquid metal plug length needs to be at least 60% of the excitation beam diameter to achieve resonant transmission greater than 50% through Port 3. The length of the plug, its range of motion, and its velocity are parameters that must be traded off, one against another, to optimize the device performance.

## Data availability

All relevant data are available from the authors upon reasonable request.

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

## Acknowledgements

The authors thank Aditya Balasubramanian for helpful discussions. This work was supported in part by the US National Science Foundation and the W.M. Keck Foundation.

## Author contributions

K.S.R. conceived the initial concept and all of the authors contributed to the design of the experiments. K.S.R., N.L.-S., R.S., and J.M. performed the measurements. K.S.R. and N.L.-S. performed the numerical simulations. I.J. and M.D.D. provided expertise on operating the liquid metals. All of the authors contributed to writing the manuscript.

## Additional information

**Competing interests:** The authors declare no competing interests.

