## [Peer Review File · Nature Communications]

Reviewers' comments:

Reviewer #1 (Remarks to the Author):

This manuscript describes the experimental and simulation results associated with an add-drop filter that has been designed to operate at THz frequencies. The device utilizes a liquid metal that can be moved about electrically, giving rise to the desired functionality. While the device is moderately interesting, there is very little that is new in what is shown. The basic notion of making a device in which the liquid metal component can be switched on reasonable timescales has been discussed in:

[1] Zhu, W. et al. A flat lens with tunable phase gradient by using random access reconfigurable metamaterial. *Adv. Mater.* 27, 4739–4743 (2015).

[2] Wu, P. C. et al. Broadband wide-angle multifunctional polarization converter via liquid-metal-based metasurface. *Adv. Opt. Mater.* 5, 1600938 (2017).

In [1], the authors used Hg, a particularly toxic material, but showed the ability to alter the device properties electrically (or, more correctly, electrically controlled pneumatic pumping). Changes in the unit cell are shown in the associated movie (supplementary information). In [2], the authors use Galinstan and HCl to cause the change in geometry. Both demonstrations were at microwave frequencies, while the current demonstration is at approximately 100 GHz. I believe that this is a detail.

In the present manuscript, the authors switch the device using an idea appears to be similar to what was detailed in co-author Dickey's previous work:

[3] Khan, M. R., Eaker, C. B., Bowden, E. F. & Dickey, M. D. Giant and switchable surface activity of liquid metal via surface oxidation. *Proc. Natl. Acad. Sci. U.S.A.* 111, 14047–14051 (2014).

[4] Khan, M. R., Trlica, C. & Dickey, M. D. Recapillarity: Electrochemically controlled capillary withdrawal of a liquid metal alloy from microchannels. *Adv. Funct. Mater.* 25, 671–678 (2014).

These also utilized chemical processes that are analogous to what was used in [2].

Thus, the present device simply puts the two ideas together, but does not show anything that is new. Under the circumstances, I cannot recommend this manuscript for publication.

Reviewer #2 (Remarks to the Author):

In this manuscript, the authors demonstrate an active "THz" (see below) channel add-drop filter based on an electrically actuated liquid metal components with passive THz waveguides. The authors claim that this device is representative of a "new design paradigm for a reconfigurable, scalable, and multi-functional approach to THz signal processing." Moreover, it is claimed that "these results pave the way for many possible reconfigurable THz devices." The device works by spatial control of the position of a liquid-metal plug. In the specific device demonstrated, the plug position determines whether THz waves propagating in one waveguide are (or are not) coupled into a second waveguide via a narrow resonance induced by the presence (or absence) of the metal plug within the beam path. On this basis, a narrow-band THz drop-add filter is demonstrated. One can imagine a range of devices based on this principle (though not demonstrated) including devices for routing, multiplexing, and demultiplexing.

The overall idea is technologically quite interesting, but this reviewer has reservations about the claims made and the possible impact of the work. Before going further, the manuscript actually presents devices functioning at ~ 100 GHz (and goes out to ~ 200 GHz). I find it something of a stretch to talk about this as a THz device. (Yes, the "THz region" is sometimes said to stretch from ~ 100 GHz to 10 THz, but this makes sense in talking about broadband time-domain systems with this sort of bandwidth. It does not make a lot of sense to call ~ 100 GHz technology terahertz technology.). The manuscript begins (abstract) by discussing the urgent need for devices enabling electrical control of processing of THz signals (though see previous comment). While acknowledging the relative lack of such devices, "urgent" is something of an exaggeration. What is more urgent in the THz field is to identify important applications in principle or in practice. Certainly, THz (or in the present case somewhat sub-THz) communications systems are of some interest; though their importance is yet to be seen, the scientific or technological significance of the device is too early to ascertain, and this topic is far from interest to the broad scientific community.

I certainly do not want to detract from the technological interest of what the authors have carried out, and there is certainly a lot of serious engineering work that has gone into their work, but I think the technological significance claimed is not compelling and the scientific significance is hard to argue in favor of. I think given the excellent work the authors are well known for, their claims actually do themselves a disservice. There is nothing wrong with discussing a drop-add filter at about 100 GHz. There is no need to claim it as a forerunner of a technological paradigm or to sell it as a THz device.

Reviewer #3 (Remarks to the Author):

In this manuscript, the authors introduce a new design paradigm for active and reconfigurable THz devices, based on the integration of electrically actuated liquid metal components and the passive waveguides in THz range, which can offer many possible applications for control of THz spectral content, wave fronts, polarization, power flow and so on. The authors demonstrate not only simulatively and experimentally the paradigm presented, but also the first channel add-drop filter for THz communications applications. More importantly, the authors show that this filter can be used to electrically switch a selected frequency channel while preserving the information in a high bit-rate modulated data stream. Therefore, it is a very interesting topic with a lot of original and full of work for readers and can be published in Nature communications.

However, the following minor changes need to be made so that readers will be more aware of the core of this paper.

1. In Section of Results, the first paragraph, the authors indicate that "We use two vertically stacked parallel-plate waveguides (PPWGs) with a gap in the shared wall between them, to allow coupling of the guided THz wave from the top to bottom waveguide. This gap is filled with one or more capillary tubes, each containing a plug of liquid metal immersed in an electrolyte solution". According to the understanding of the reviewer, the authors describe the process of THz wave coupling to the bottom waveguide from the top waveguide and optimize various parameters of the paradigm in the Supplementary Information. The authors should clearly explain the Physical cause.
2. In Section of Results, the second paragraph, the authors describe "The capillary is connected to the voltage source via platinum wires placed either directly inside the ends of the filled capillary tube or into reservoirs located at either end". Obviously, it is an important part of the experimental work for the voltage-actuated capillary containing the liquid metal. However, the reviewer could not find the impact of the voltage on the characterization of the filter from the figure 2 to the figure 5. Please give the illustration.

Response to referee comments

Reviewer #1 (Remarks to the Author):

This manuscript describes the experimental and simulation results associated with an add-drop filter that has been designed to operate at THz frequencies. The device utilizes a liquid metal that can be moved about electrically, giving rise to the desired functionality. While the device is moderately interesting, there is very little that is new in what is shown. The basic notion of making a device in which the liquid metal component can be switched on reasonable timescales has been discussed in:

[1] Zhu, W. et al. A flat lens with tunable phase gradient by using random access reconfigurable metamaterial. Adv. Mater. 27, 4739–4743 (2015).

[2] Wu, P. C. et al. Broadband wide-angle multifunctional polarization converter via liquid-metal-based metasurface. Adv. Opt. Mater. 5, 1600938 (2017).

In [1], the authors used Hg, a particularly toxic material, but showed the ability to alter the device properties electrically (or, more correctly, electrically controlled pneumatic pumping). Changes in the unit cell are shown in the associated movie (supplementary information). In [2], the authors use Galinstan and HCl to cause the change in geometry. Both demonstrations were at microwave frequencies, while the current demonstration is at approximately 100 GHz. I believe that this is a detail.

In the present manuscript, the authors switch the device using an idea appears to be similar to what was detailed in co-author Dickey's previous work:

[3] Khan, M. R., Eaker, C. B., Bowden, E. F. & Dickey, M. D. Giant and switchable surface activity of liquid metal via surface oxidation. Proc. Natl. Acad. Sci. U.S.A. 111, 14047–14051 (2014).

[4] Khan, M. R., Trlica, C. & Dickey, M. D. Recapillarity: Electrochemically controlled capillary withdrawal of a liquid metal alloy from microchannels. Adv. Funct. Mater. 25, 671–678 (2014).

These also utilized chemical processes that are analogous to what was used in [2].

Thus, the present device simply puts the two ideas together, but does not show anything that is new. Under the circumstances, I cannot recommend this manuscript for publication.

We feel that this referee has failed to recognize the vastly greater challenges associated with the construction of devices for signals above 100 GHz, compared to the fabrication of devices for lower frequencies. It is simply not true that operation of devices at 100 GHz is merely a detail. Entirely new approaches are required, even when scaling from 5 GHz to 60 GHz (let alone 5 GHz to 100 GHz). The evidence for this is abundantly clear, for example in all of the protocols for soon-to-be-released 5G wireless networks. The low-frequency 5G bands and the millimeter-

wave 5G bands are completely different beasts, in almost every respect. To conflate these and describe the distinction as “a detail” is incorrect and misleading.

To be fair, the referee does have some valid points. The two metamaterial references are significant works that we should have referenced, so **we have modified our manuscript to include these two references**. Also, it is true that we have not pioneered a new method for electrically actuating a liquid metal – of course, we never claimed to have done so. In earlier work, Dickey’s team reported this general approach in several papers (a few of which we had already cited, including one of the two mentioned by the referee). However, we note that, in those earlier works, the composition of the electrolytic fluid was not optimized for its terahertz optical properties. This is a very important aspect of our work: our demonstrated devices would not work at all if we had not first undertaken this crucial optimization step (discussed in fig. S1 of the Supplementary information). Indeed, prior to beginning this project, it was not at all obvious that we would find ANY formulation of the electrolyte that would work. This is because lowering the THz absorption (e.g., by replacing water with methanol) also lowers the DC electrical conductivity, which diminishes the effectiveness of the electrical switching of the liquid metal plug. Therefore, we cannot agree with the referee’s assertion that nothing new is shown here. However, we do agree that we did not emphasize the importance of this electrolyte optimization sufficiently. So **we have modified the text of the manuscript in order to make this point clearer**.

Reviewer #2 (Remarks to the Author):

In this manuscript, the authors demonstrate an active "THz" (see below) channel add-drop filter based on an electrically actuated liquid metal components with passive THz waveguides. The authors claim that this device is representative of a “new design paradigm for a reconfigurable, scalable, and multi-functional approach to THz signal processing.” Moreover, it is claimed that “these results pave the way for many possible reconfigurable THz devices.” □The device works by spatial control of the position of a liquid-metal plug. In the specific device demonstrated, the plug position determines whether THz waves propagating in one waveguide are (or are not) coupled into a second waveguide via a narrow resonance induced by the presence (or absence) of the metal plug within the beam path. On this basis, a narrow-band THz drop-add filter is demonstrated. One can imagine a range of devices based on this principle (though not demonstrated) including devices for routing, multiplexing, and demultiplexing.

The overall idea is technologically quite interesting, but this reviewer has reservations about the claims made and the possible impact of the work. Before going further, the manuscript actually presents devices functioning at ~100 GHz (and goes out to ~200 GHz). I find it something of a stretch to talk about this as a THz device. (Yes, the “THz region” is sometimes said to stretch from ~100 GHz to 10 THz, but this makes sense in talking about broadband time-domain systems with this sort of bandwidth. It does not make a lot of sense to call ~100 GHz technology terahertz technology.)

We must respectfully disagree with the referee here. By now, it is standard in the literature to refer to signals above 100 GHz using the terminology that we've used. See, for example these recent articles, written by three of the leading research groups in the field:

<https://www.sciencedirect.com/science/article/pii/S1874490714000238>

Entitled “**Terahertz** band: next frontier for wireless communications,” the first sentence of the abstract reads “This paper provides an in-depth view of terahertz band (0.1-10 THz) communication...”

<https://link.springer.com/article/10.1007/s10762-010-9758-1#Sec13>

Entitled “A review on **terahertz** communications research,” and containing an entire section devoted to 120 GHz data links.

<http://ieeexplore.ieee.org/stamp/stamp.jsp?arnumber=6892933>

Entitled “**Terahertz** photonics for wireless communications,” and containing a sentence in the introduction which says “At terahertz frequencies (>100 GHz), not only is the free space path loss higher,...”

These articles all use the word “terahertz” and define it as “greater than 100 GHz”, and all in the context of wireless communications (i.e., they are not talking about broadband time-domain sources). And this is hardly a complete list – it would be easy to provide links to many more articles which use the same terminology in the same way. Indeed, the most highly cited article in the history of the IEEE THz journal is entitled “Present and future of terahertz communications,” (Song and Nagatsuma, IEEE Trans. THz. Sci. Technol. 1, 256-263 (2011)) and mostly focuses on devices and systems operating in the 100-300 GHz range (although noting that frequencies above 275 GHz are of particular interest because they have not yet been allocated to specific users by the relevant regulatory agencies). The referee may find it dubious to refer to devices that operate above 100 GHz as “terahertz devices”, but the field clearly disagrees.

To more clearly emphasize what we mean in our discussion, **we have modified the language of our first paragraph** in order to clarify that “terahertz” means “above 100 GHz”.

The manuscript begins (abstract) by discussing the urgent need for devices enabling electrical control of processing of THz signals (though see previous comment). While acknowledging the relative lack of such devices, “urgent” is something of an exaggeration. What is more urgent in the THz field is to identify important applications in principle or in practice.

This is another minor quibble, but once again we are compelled to respectfully disagree. The field has **already** identified important applications, both in principle and in practice, including (but certainly not limited to) wireless communications. It is unfortunate that the referee is unaware of the many applications that are already in commercial use, let alone the many that are under active research and development. Whether or not the particular need that we've addressed is ‘urgent’ is a matter of perspective, I suppose. One cannot envision a terahertz wireless system until the components are developed, but one cannot find the motivation to develop them unless the need has risen to a certain level of urgency. The recent heightened activity in the field seems to suggest that a certain level of urgency has been reached by now (see, e.g., the \$27 million

multi-university multi-year center awarded in February 2018 to a team led by Mark Rodwell from UCSB, devoted very specifically to the development of THz communication devices). Yet, our disagreement notwithstanding, it is certainly not our intention to engage in unwarranted hyperbole. Therefore, **we have edited the text of our manuscript to tone down the language**, removing the word “urgently” and no longer referring to a ‘new paradigm’. We trust that these modifications will address the concern.

Certainly, THz (or in the present case somewhat sub-THz) communications systems are of some interest; though their importance is yet to be seen, the scientific or technological significance of the device is too early to ascertain, and this topic is far from interest to the broad scientific community.

We once again disagree with the referee’s assessment of broad interest. Given the large (and rapidly growing) number of research groups in the field of terahertz communications, it seems clear that an innovation in the field would be of broad interest. However, we are sensitive to the critique that our one device demonstration (an add-drop filter) may not fully communicate the versatility and power of our general approach. Therefore, **we have added a significant set of new data to our manuscript**. These new results describe a second (completely different) device, also based on the same general approach of integrating liquid metals with passive waveguide components, providing another unique new functionality for terahertz signal processing. Note that the electrolyte used to switch the liquid metal’s position in this new device may need to be different from that used in our add-drop filter, depending on the add-drop filter’s input wave polarization (which changes the mode of coupling between the two waveguides). This new result is not merely a rehash of what we already did.

I certainly do not want to detract from the technological interest of what the authors have carried out, and there is certainly a lot of serious engineering work that has gone into their work, but I think the technological significance claimed is not compelling and the scientific significance is hard to argue in favor of. I think given the excellent work the authors are well known for, their claims actually do themselves a disservice. There is nothing wrong with discussing a drop-add filter at about 100 GHz. There is no need to claim it as a forerunner of a technological paradigm or to sell it as a THz device.

As noted in our comments above, the revised manuscript now describes two different devices based on the general approach, which enable two different signal processing functions for terahertz waves. We feel that it now emphasizes more clearly the new approach here, and we hope that the referee agrees with this assessment. Also, as noted above, yes, these are THz devices.

Reviewer #3 (Remarks to the Author):

In this manuscript, the authors introduce a new design paradigm for active and reconfigurable THz devices, based on the integration of electrically actuated liquid metal components and the passive waveguides in THz range, which can offer many possible applications for control of THz spectral content, wave fronts, polarization, power flow and so on. The authors demonstrate not only simulatively and experimentally the paradigm presented, but also the first channel add-drop

filter for THz communications applications. More importantly, the authors show that this filter can be used to electrically switch a selected frequency channel while preserving the information in a high bit-rate modulated data stream. Therefore, it is a very interesting topic with a lot of original and full of work for readers and can be published in Nature communications.

We appreciate the positive remarks of the referee, especially the recognition of the novelty and originality of our work.

However, the following minor changes need to be made so that readers will be more aware of the core of this paper.

1. In Section of Results, the first paragraph, the authors indicate that “We use two vertically stacked parallel-plate waveguides (PPWGs) with a gap in the shared wall between them, to allow coupling of the guided THz wave from the top to bottom waveguide. This gap is filled with one or more capillary tubes, each containing a plug of liquid metal immersed in an electrolyte solution”. According to the understanding of the reviewer, the authors describe the process of THz wave coupling to the bottom waveguide from the top waveguide and optimize various parameters of the paradigm in the Supplementary Information. The authors should clearly explain the Physical cause.

We are not entirely sure that we understand this comment; we think that the referee is asking about the physical mechanism which underlies the coupling between the two waveguides. If so, then the simulation in the original figure 1 (now fig. 2) should provide at least a partial answer. When the capillary is filled with metal (as in Fig. 2d), the gap between the waveguides is defined by the capillary walls, through which strong coupling can occur (the capillary is glass, and only weakly absorbing). In contrast, when the capillary is filled with liquid electrolyte (Fig. 2c), the effective gap size (and transmission loss) is very different, so the coupling is strongly inhibited at the relevant frequency. Aside from the complexity associated with the not-so-simple shapes of the components, this is a fairly straightforward problem in resonant coupling between two waveguides through a wavelength-scale gap. To further clarify this question, **we have added some additional information to the Supplementary materials.**

2. In Section of Results, the second paragraph, the authors describe “The capillary is connected to the voltage source via platinum wires placed either directly inside the ends of the filled capillary tube or into reservoirs located at either end”. Obviously, it is an important part of the experimental work for the voltage-actuated capillary containing the liquid metal. However, the reviewer could not find the impact of the voltage on the characterization of the filter from the figure 2 to the figure 5. Please give the illustration.

The effect of the voltage is to move the metal plug in and out of the interaction region of the coupling gap between the two waveguides, within the capillary tube. In essence, we think of this as a binary system – either the plug is “in” or it is “out”. Thus, there are only two voltage values that matter: the one that causes the plug to be inserted into the active region of the waveguide gap, and the one that causes it to be withdrawn. An intermediate value between these two voltages would cause the plug to be ‘half-way in’, which would lead to partial electromagnetic coupling between the waveguides. We did not illustrate this behavior in the manuscript because it is not clear that this ‘half-way’ configuration would be useful for anything. In the context of an

add-drop filter, one wants the signal to emerge from one port or the other, not both. An investigation of the details of the intermediate cases would therefore seem to be more suitable for a subsequent publication.

REVIEWERS' COMMENTS:

Reviewer #2 (Remarks to the Author):

The authors have addressed both the major and minor points I have raised in my past review. And while I will not push my point about terminology (i.e., just what THz refers to; I am well aware of how this term is widely misused) and indeed the authors have made adequate revisions in their manuscript concerning the minor points I raised, I stand by my comments concerning the technological and scientific importance of the work.

I am well aware of applications of THz in the real world; however, much of the "real-world" interest cited has more to do with major research areas than with actual applications--what companies are or are likely to spend their money on. We are talking about a Nature journal: the onus is on the authors to argue the major scientific impact of the work or the ground-breaking progress they have made in solving a well-defined and significant technological problem that is specifically holding up major new developments. where there are users champing at the bit. All the major applications cited really face numerous technological challenges, and given the timescale likely for widespread deployment of THz-based applications, it is hard to argue that the authors' drop-add filter is the way these systems will go. Again, I have no problem with the authors' work; I just do not see that the work merits publication in a Nature journal.

Reviewer #4 (Remarks to the Author):

This work presents an electrically actuated device for THz signal processing. Demonstration of robust add-drop functionality is presented at 123 GHz, scalable up to 200 GHz. This is well within the accepted THz band and relevant to next generation wireless communications, and therefore of interest to the optical communications field and terahertz communities at large. The work is well presented with all claims supported by experimental data and finite element simulations. I believe this to be a very nice feat of engineering, but also an important step towards functional communication devices in the THz band. The urgency for the need of such devices is debatable perhaps, however to me the push to higher frequencies is inevitable and the need for such technologies will be there eventually. Major communications centers, beyond those that the author refers to in their response, are gearing up for a push towards THz frequencies. This is the new frontier for communications, that is clear.

In my opinion, the concerns of the previous reviewers have been addressed. Many of these concerns were based on misconceptions such as where to draw the line between GHz and THz. This is not so important in my opinion. What is important is that to move from established GHz technology to 100 GHz and definitely 200 GHz, one needs to rethink the entire component platform. This to me defines a new spectral regime for communications and should therefore be called THz, in agreement with the rest of the community as the authors point out. The extension to other components would be very interesting, particularly wave front engineering of guided THz light that can be electrically controlled. While this is beyond the scope of this paper perhaps, it seems reasonable that such capabilities are feasible within this platform. Regardless, the current demonstration is of sufficient interest to merit publication.

In summary, I endorse this paper for publication in Nature Communications.

Reviewer #5 (Remarks to the Author):

This paper presents simulation and experimental results of two example THz devices developed by integrating liquid metal components into THz waveguides. This new approach enables active, rapid

and flexible control of devices and there is scope for many other devices being developed based on this technology. The devices presented, namely an active power splitting switch, and channel add-drop filter are the first of their kind.

The rationale behind the device development and the potential for other devices to be designed is clearly explained and it is clear to me that this is a very important step for THz communications. While building blocks of this idea have been previously published, and duly cited, the breakthroughs needed in order to put all the aspects together to make the technology work for 0.1 THz are presented in this paper.

Developments at 0.1 THz are important to the THz community as well as wider communications fields. It is very likely that THz communications will become mainstream in the near future and so developing such devices to support this is paramount. It is splitting hairs to debate whether we should class this work as GHz or THz – it is excellent work and deserves to be published in Nature communications.